# Levels of Neutralizing Antibodies against SARS-CoV-2 in Mothers and Neonates after Vaccination during Pregnancy

**DOI:** 10.3390/vaccines11030620

**Published:** 2023-03-09

**Authors:** Antônio Oliveira da Silva Filho, Daniel Nazário Gonçalves, Letícia Karina Ramos de Lima, Dâmocles Aurélio Nascimento da Silva Alves, Felipe José de Andrade Falcão, Rosângela Estevão Alves Falcão

**Affiliations:** 1Post-Graduate Program in Health and Socio-Environmental Development Multicampi Garanhuns, University of Pernambuco, Garanhuns 55294-310, Brazil; 2Department of Health Sciences, Multicampi Garanhuns, University of Pernambuco, Garanhuns 55294-310, Brazil; 3Department of Exact Sciences, Multicampi Garanhuns, University of Pernambuco, Garanhuns 55294-310, Brazil

**Keywords:** COVID-19, maternal immunization, neutralizing antibodies, SARS-CoV-2, BNT162b2

## Abstract

Background: Maternal vaccination is safe and provides protection against COVID-19 in mothers and neonates, and it is necessary to evaluate its effect on the induction of immune responses through the levels of neutralizing antibodies present in maternal and neonatal blood. Methods: An observational study with transversal analysis was carried out. Included in the research were neonates (<1 month) whose mothers had been immunized whilst pregnant with at least one dose of the vaccine BNT16b and had not shown any symptoms of COVID-19. The blood of the mothers and newborns was collected during the Guthrie test and sent to the laboratory for the detection of neutralizing antibodies against SARS-CoV-2. Results: A total of 162 pairs of mothers and neonates were analyzed with an average age of 26.3 ± 5.97 years and 13.4 ± 6.90 days, respectively. All samples collected present neutralizing antibodies with an average percentage of 91% in the mothers and 92% in the neonates. The most satisfactory immune response was observed in neonates and mothers vaccinated during the second trimester of gestation. Conclusions: The vaccination of expectant mothers with the immunizer BNT162b2 has promoted a robust immunological response in both the mothers and the neonates.

## 1. Introduction

During 2020, the World Health Organization (WHO) decreed the coronavirus disease as a pandemic (COVID-19) caused by the coronavirus-2 associated with acute respiratory distress syndrome (SARS-CoV-2) [1]. It is known that expectant mothers are at a higher risk of developing both severe clinical presentations of COVID-19 and maternal and perinatal complications due to the physiological alterations that occur during the gestational time period and are therefore placed in priority at-risk groups for immunization [2,3].

With the rapid spread of the pandemic, global efforts were made with the intent to develop efficient vaccines in a short time span [4]. Among them were stand-out vaccines, such as the messenger RNA (mRNA) BNT162b2 vaccine developed by Pfizer [5]. Given the concerns over their safety and possible unknown side effects, expectant mothers were excluded from the testing performed to ensure the efficiency of the immunizers [2,6]. However, in the face of the risk of complications of a possible infection, several medical entities, such as the Center for Disease Control (CDC) and the American College of Obstetricians and Gynecologists (ACOG), recommended maternal vaccination [4].

Currently, the evidence shows that the immunization of expectant mothers with mRNA vaccines such as BNT162b2 is safe, and in addition to protecting mothers, they are associated with the development of a neonatal immunological response through the possibility of passive transmission, supported by the presence of specific antibodies against SARS-CoV-2 present in breast milk, maternal serum, the placenta, and the umbilical cord’s blood [7]. Data from current research show that the vaccine produced by Pfizer induces high levels of neutralizing antibodies in the maternal blood as well as neonatal blood [5] and breast milk [8], confirming its efficiency.

However, there are still questions to be answered due to the lack of literature on various topics, for example, concerning the gestational time point when vaccination will promote a stronger immunological response in the mother as well as in the neonate [5] and aid in the durability of the child’s immunological protection as time goes on [4]. In this context, this work is focused on identifying mother–child transmissions of neutralizing antibodies (NABs) through their presence in the blood of neonates and their vaccinated mothers who received at least one dose of BNT162b2, with the goal of evaluating neonatal immunological protection induced by vaccination and to observe at which point during gestation immunization is able to generate a strong immunological response in neonates.

## 2. Materials and Methods

### 2.1. Study Blueprint

This study comprises an observational study, with transversal analysis, classified as field research of an applied quantitative nature.

### 2.2. Sample and Population

The study was conducted in the town of Garanhuns, a town with 141, 347 inhabitants situated in the region of Meridional Agreste, 232 km away from the capital, Recife. The collection of data and biological samples was carried out in the children’s hospital Palmira Sales, a health center and philanthropic organization. Included in the research were neonates (<1 month) whose mothers had been immunized while pregnant with at least one dose of the vaccine BNT16b2 (Pfizer, New York, NY, USA). Mothers who had shown symptoms of COVID-19 were not suitable for this research, as were preterm neonates and children that had shown symptoms of the disease.

### 2.3. Collection of Biological Material

Biological material (blood) from the neonates was obtained during the Guthrie’s test procedure, which was carried out during the 28 days after birth, where 2 mL of blood was collected with a needle attached to a 3 mL syringe. Maternal blood was obtained following the same specifications. The samples were then placed inside a specific cooling unit for transportation, with the temperature being monitored by a thermometer in order to maintain a temperature of between 2 and 8 °C, and were later sent to the laboratory headquarters, ALFALAB, where the neonates’ test registration and the processing of samples for analysis took place.

The detection of neutralizing antibodies was carried out through fluorescent immunoassays (FIA) for the quantification of NABs against SARS-CoV-2 using the Boditech Med Incorporated kit. The results were classified as positive or negative according to the cut-off index (COI), which is found by analyzing the interference of the protein spike RBD from SARS-CoV-2 with the receptor ECA-2 through the presence of NABs. If the result is equal or superior to 30%, the test is considered positive regarding the presence of NABs. [9].

### 2.4. Data Analysis

After triage and the collection and processing of biological material, the data obtained were analyzed descriptively through absolute distribution and percentages using the software Microsoft Excel 2013, E.U.A For the correlation of the dependent and independent variables, a Pearson’s test with a posterior construction of the graph through multiple linear regressions was carried out. The F test was utilized to verify the independent variables, and the T test was used to investigate the correlation between the average maternal and neonatal antibodies in relation to the gestational month of vaccination; results were considered significant when *p* > 0.05. To investigate at which gestational time point maternal vaccination is able to induce a robust immunological response in the neonate, Tukey’s test was carried out, comparing all average pairs based on the minimum substantial difference (D.M.S.). All statistical analyses were carried out using the software RStudio 2022.02, E..U.A. 

### 2.5. Ethical Considerations

The study was approved by the Ethics Council of Research Multicampi Garanhuns University of Pernambuco under statement nº 5.281.016. The collection of data and biological material only took place after the signing of the Terms of Free Informed Consent (Termo de Consentimento Livre Esclarecido TCLE) by the mother/individual responsible for the child.

## 3. Results

Between April and October of 2022, 172 samples were collected from women who were vaccinated with BNT162b2 during the gestational period. A total of 172 samples were also collected from the newborns, resulting in a total of 172 pairs. Of these, ten were removed as they did not meet the selection criteria (four were preterm neonates, and six were over 28 days old at the time of collection), resulting in the analysis of 162 pairs in this study. Respectively, the average ages of the mothers and neonates were 26.3 ± 5.97 years and 13.4 ± 6.90 days, respectively, with an average gestational age of 39.19 ± 1.16 weeks. The mother and neonate’s data are given in Table 1.

Women under 30 years of age and neonates under 14 days during sample collection were analyzed in higher numbers. The women vaccinated during their first gestational trimester were the majority (105/162), representing 64.81% of the total. Regarding the first dose of the vaccine during pregnancy, in 87 (53.70%) of the analyzed mothers, the first dose received during pregnancy was also their first dose of BNT162b2; in 49 (30.2%) mothers, the first dose during pregnancy was their second dose of the vaccine; and in 26 (16.05%) mothers, the first dose during pregnancy was the booster dose. Regarding the feeding of the neonate, 69.13% were exclusively breastfed until the point of data collection.

The average amount of antibodies in neonates whose mothers received the first dose during gestation (87 mothers) corresponds to 90%; of those, four did not receive their second dose, with an average of 94% of antibodies in the mothers and 88% in the neonates. A total of 66 mothers received two doses, with an antibody average of 89% in mothers and 90% in neonates; eleven received three doses—including the booster dose—with an antibody average of 95% in mothers and 96% in neonates; and six provided no information. From the information above, we can say that in our data, mothers who were vaccinated with a single dose during pregnancy passed on a lower amount of antibodies to their babies than mothers who received two doses, and the same can be seen if we compare mothers who received two doses with those who received three doses.

In mothers whose first dose during pregnancy was their second (49 mothers), newborns had an antibody average of 94%, and of these mothers, 24 received the booster dose during pregnancy (i.e., two doses while pregnant), and 25 did not. Of the 25 who did not receive the booster, both newborns and mothers had an average of 94% neutralizing antibodies, and the same occurred with the 24 mothers who received the second dose of the vaccine. Finally, in mothers whose first dose during gestation was a booster dose and who therefore received only one dose during gestation (26 mothers), the mothers’ antibody average was 92%, and the neonates’ was 94%.

Specific neutralizing antibodies against COVID-19 were detected in all maternal and neonatal blood samples, with an average percentage of 91% (10 to 99%) in mothers and 92% in neonates (8–99%). A total of 138 mothers (85.19%) and 141 neonates (87.04%) presented a level of neutralizing antibodies equal or superior to 90% by means of the multiple linear regression. It was possible to correlate the antibody average in neonates and mothers to the month of vaccination during the gestational period, as illustrated in Table 2.

The existing relationship between the percentage of neutralizing maternal and neonatal antibodies was observed, resulting in a Pearson’s correlation coefficient equal to 0.95 and demonstrating that as the average maternal antibody level increases, the average antibody level of the neonates also tends to increase (Figure 1).

In the multiple linear regression model, all assumptions of reliability were met, indicating that it can be used to predict the average level of antibodies in the neonate from the maternal antibodies and gestational month of the first dose during pregnancy. The calculated R2 (coefficient of determination) was equal to 0.935, and the adjusted R2 was equal to 0.9133 (indicating a high explanation of the model in the dependent variables); thus, the month in which the mother was vaccinated and her level of neutralizing antibodies explain 93.5% of the variability of the neonate antibodies. However, there was no statistically significant relevance regarding the month in which the mother received the vaccination, at *p* = 0.201, which means that, according to the results of this test, there is no way to predict/relate the best month for the vaccination of mothers to ensure that the neonate obtains a higher percentage of antibodies. However, the *p*-value of the average maternal antibody variable was 0.000106 < 0.05, which means that, for every 1% of the mother’s antibody level percentage, the percentage of the neonate increases by an average of 0.78%.

Using an ANOVA test, it was possible to demonstrate that there is no difference in the production of antibodies in the mother when compared to the neonate; however, a difference was noticed in the production of antibodies in the neonate according to the month in which the mother was vaccinated (*p* = 0.0000283). Beyond that, the value of R2 = 0.9133 indicates that the month in which the mother was vaccinated and her level of neutralizing antibodies explains 93.5% of the variability of antibodies in the neonate. That is to say, a neonate whose mother was vaccinated during the first 6 months of gestation and presents 95% neutralizing antibodies will have, on average, a rate of antibodies equal to 74.3%.

In order to obtain answers regarding the best period for vaccination during pregnancy with the aim of achieving higher percentages of neutralizing antibodies in neonates, an ANOVA test was also performed. Through this test, it was identified that there is no difference in the antibody production of the mother when compared to the neonate; however, there is a difference in antibody production by the neonate, depending on the month in which the mother was vaccinated (*p* = 0.0000283).

The aforementioned test informs us that there is a time point during gestation where maternal vaccination induces a more robust immune response in the neonate; however, for this to be elucidated, Tukey’s test was performed. In this test, a comparison was made between the mean antibodies of the newborn (Table 2) month by month in order to uncover in which month the newborn statistically expresses a higher percentage of antibodies. The same was carried out with mean maternal antibodies. Thus, in order to choose the best month for vaccination, it is useful to examine the table below; the first and fifth columns give the compared months, where it can be seen that there are statistically significant differences, and those whose values of minimum significant differences are greater than 0.00393 (Table 3) are shown in light gray. To verify which month has a better performance, we observe, among the highlighted comparisons, the averages arranged in columns 2 and 3 in relation to the newborn’s antibodies and in columns 6 and 7 in relation to the mother’s. These averages are taken from Table 2. Overall, maternal vaccination was less effective in the third trimester.

It is possible to observe that if the vaccine is received in the first or third month, for example, the average level of antibodies in the neonate changes, with the first month being the best month for the production of antibodies because its average is higher. The same is true when comparing the second, fourth, sixth, and ninth months with the third, where the latter will always present a lower average in relation to the others. In the case of mothers, the same situation occurs when comparing the average antibodies of mothers vaccinated in the seventh month with those of the first, second, fourth, fifth, sixth, eighth, and ninth months. The seventh month assumes lower average antibodies; therefore, it is suggested that it would be better to receive the vaccine during the other months in cases of possible choice.

It should be highlighted that our study showed that the maternal and neonatal immunological response was generally more efficient in children and mothers vaccinated in the second trimester of gestation, with an average percentage of antibodies of 94% and 93%, respectively.

## 4. Discussion

In the present study, the vaccination of pregnant women with the immunizer BNT162b2 promoted a robust immune response in both mothers and neonates, with high levels of neutralizing antibodies detectable in maternal and neonatal blood regardless of the gestational time point at which the immunization process occurred, corroborating the confirmation of the benefits of maternal immunization for protection against SARS-CoV-2.

It was observed in our research that all mothers and neonates presented neutralizing antibodies against SARS-CoV-2. Corroborating these findings, in Popescu et al. (2022), the presence of anti-RBD was detected in the maternal blood and in the neonate’s umbilical cord in all 91 evaluated pairs [10]. In the study by Nir et al. (2022), the presence of antibodies was detected in the blood from the umbilical cord at levels of 96.4% in neonates and 100% in the maternal blood [11]. In addition, Rottenstreich et al. (2022) identified the presence of specific anti-RBD antibodies in all 228 neonates of mothers vaccinated with BNT162b2 [12].

In our study, it was not possible to assess whether the antibodies in the neonates come from the transfer at delivery or from breastfeeding; however, we saw that even in cases where the collection was carried out in the first days of the child’s life, antibody levels were high, pointing to a transfer of antibodies from the mother to the baby. Recent research confirms both possibilities of the passive transmission of immunity against COVID-19. Kugelman et al. (2022) detected the presence of neutralizing antibodies in maternal and neonatal blood collected at the time of delivery from 102 patients vaccinated with the BNT162b2 booster dose and in 93 newborns [13].

Kashani-Ligumsky et al. (2021) compared the antibody levels in three groups: 1. neonates and mothers infected by COVID-19; 2. neonates of mothers vaccinated with two doses of BNT162b2 during the third trimester of gestation; and 3. neonates and mothers who were not vaccinated or infected. It was observed that the average specific antibody level in the samples of blood and umbilical cords was significantly higher in the newborns of vaccinated women than in those of mothers who contracted SARS-CoV-2 during pregnancy—224.7 ± 64.3 u/mL vs. 83.7 ± 91.6 u/mL (*p* < 0.05) [14]. Our study, as stated earlier, dealt only with cases in which the mothers had shown no symptoms of COVID-19, and it is therefore suggested that neonatal immunization in the study population comes mostly from vaccination or breastfeeding.

Regarding breastfeeding, several studies have already shown the presence of immunoglobulins against SARS-CoV-2 in breast milk. In the study by Ricciardi et al. (2022), involving 18 infants vaccinated with at least one dose of BNT162b2, breast milk samples were collected at different times after vaccination and analyzed at points before the first dose, two weeks after the second dose and six months after the first dose. The presence of antibodies was detected in all samples analyzed, with a decrease after 6 months post-vaccination [8].

Regarding this issue, in the present study, we observed that 146 of the 162 newborns evaluated were breastfeeding until the time of collection, 112 of them exclusively. It is known that breastfeeding, aside from providing the necessary nutrients for the child’s healthy development and the strengthening of the bond between mother and child [1], contributes significantly to the neonate’s humoral immune response through the passage of antibodies [15]. Even so, the neonates who were only formula-fed obtained, in all six observed cases, very similar percentages of neutralizing antibodies to those of the mother, which points to the high efficacy of antibody transfer via the vaccine, even without breastfeeding, in the child’s first month of life.

The recent results confirm that the vaccination against SARS-CoV-2 during the gestational period is not only safe but also provides protection for pregnant women and their children, with detectable maternal antibodies found in the blood of the umbilical cord, breast milk, and serum samples obtained from the infants. Findings indicate the transference of maternal antibodies, whether through the transplacental route or breastfeeding [7,11]. In general, the antibody concentrations found in the blood of neonatal umbilical cords of vaccinated mothers are strongly correlated with maternal antibody levels [16].

In our study, we saw that the existing relationship between maternal antibodies and neonate antibodies is high, and we observed that the values are visually very similar, which is numerically confirmed through the Pearson’s correlation coefficient between the two variables, corresponding to 0.95; thus, as the mean maternal antibodies grow, the mean neonate antibodies also tend to grow, and furthermore, maternal antibodies account for about 95% of the variation in neonate antibodies.

Such correlation was also observed in our research, corroborating results previously described in the literature. In this comparative study of vaccination effects and maternal infection by COVID-19 in terms of the immunological response, Nir et al. (2022) reported the efficient transmission of specific anti-SARS-CoV-2 antibodies in the 64 women that received the Pfizer mRNA vaccine during pregnancy, with a positive correlation between the maternal serum concentration and antibodies in the blood from the umbilical cord (r = 0.483; *p* = 0.0001) [11].

It is noteworthy that, in our study, the data were collected over 7 months, which included the dose of the vaccine that the mother received during pregnancy. As time passed, the women became pregnant after having received one or more doses, but all received at least one dose during pregnancy, whether it was the first, second, booster, or even two or three of these. 

One of the major dilemmas regarding maternal vaccination is the ideal time point, that is to say, which week/month/trimester will provide a stronger neonatal immunological response. In our study, it was observed that during the first and third trimesters of gestation, vaccinated mothers and their neonates presented an average rate of antibodies (91% in both) that was a little inferior when compared to newborns immunized during the second trimester (94%). Such results go against the majority of findings described in the literature.

Yang et al. (2022) examined 1359 women that received at least one dose of both the mRNA (Pfizer, Moderna) and of the viral vector (Jansen) and their 1374 neonates, and it was generally observed that immunization during any period of pregnancy is associated with detectable levels of maternal antibodies at birth; however, higher levels of maternal antibodies and, consequently, antibodies in the blood from the umbilical cord, occur from vaccination in the third trimester [17].

Our study has provided data that differ from some of the research described in the literature in regard to the most suitable time point for vaccination during gestation. This could be a consequence of the higher rate of women vaccinated in the first trimester (64.1%) and significant disproportionality in relation to the second and third trimesters (only 6.79% were vaccinated in the third trimester). This was due to the long period over which the collection was carried out (7 months) to achieve a good number of pairs (mother/infant) for analysis. Furthermore, a larger number of pairs is essential for analysis and is better able to more precisely define the best gestational time point that will provide a stronger immunological response against SARS-CoV-2 in the neonate.

However, it is important to emphasize that studies show that mothers vaccinated at any point during gestation present neutralizing antibody levels that are satisfactory and can guarantee the protection of the neonates through transmission via the placenta or breastfeeding [17,18]. Furthermore, maternal vaccination contributes significantly to minimizing infection risks in severe cases of acute respiratory syndrome in expectant mothers [19].

Current data suggest that neonatal protection generated by maternal immunization by two immunizing doses of an mRNA vaccine, such as BNT162b2, is also efficient in decreasing the hospitalizations caused by COVID-19 in infants aged under six months infected with SARS-CoV-2 [7]. Thus, the immunization of expectant mothers is beneficial and should be incentivized to promote maternal and neonatal protection against COVID-19.

## 5. Conclusions

High rates of neutralizing antibodies are detected in the maternal and neonatal blood independently from the gestation trimester when immunization occurs, the benefits of which can be transmitted passively, either transplacentally or through breastfeeding, thus contributing to the protection of the neonate. More studies are necessary to further understand the durability and kinetics of these antibodies in the long term, as well as their effectiveness against new variants of SARS-CoV-2.

## Figures and Tables

**Figure 1 vaccines-11-00620-f001:**
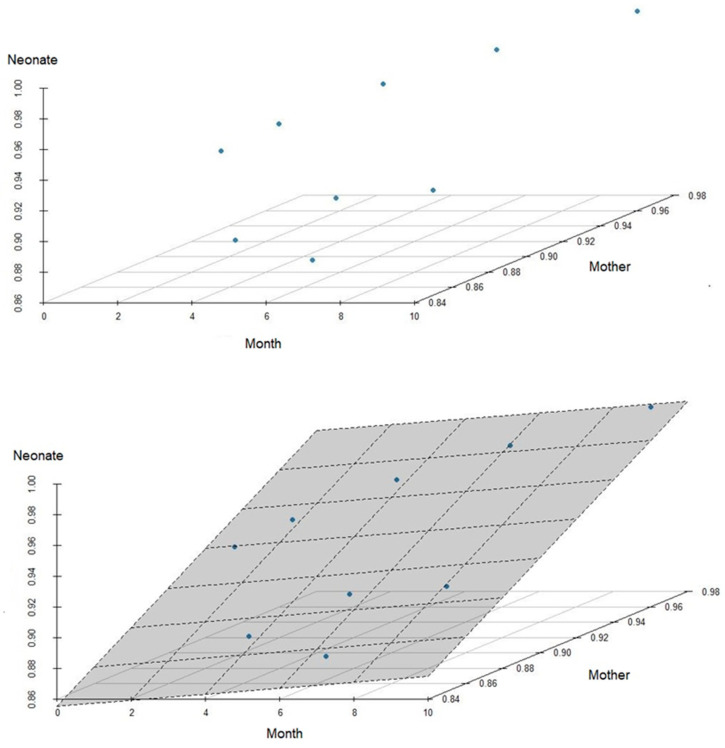
Graph of dispersion of the correlation between the percentage of maternal and neonatal antibodies.

**Table 1 vaccines-11-00620-t001:** Data from the mothers and neonates.

Parameters	N = 162 (100.00%)
Mother’s age (in years)	
15–20	29 (17.90%)
21–25	51 (31.48%)
26–30	38 (23.46%)
31–35	33 (20.37%)
≥36	11 (6.79%)
Neonate’s age (in days)	
1–7 days	50 (30.87%)
8–14 days	61 (37.65%)
15–21 days	28 (17.28%)
22 and 28 days	23 (14.20%)
Gestational period during vaccination	
1st Trimester	105 (64.81%)
2nd Trimester	46 (28.40%)
3rd Trimester	11 (6.79%)
Dose received during gestation	
1st Dose	87 (53.70%)
2nd Dose	49 (30.25%)
Reinforcement dose	26 (16.05%)
Type of feeding for the neonate	
Exclusively breastfeeding	112 (69.13%)
Breastfeeding + complement	34 (20.99%)
Only complement	5 (3.09%)
Data not given	11 (6.79%)
Neutralizing antibodies (%) in mothers	
1–29%	2 (1.23%)
30–59%	7 (4.32%)
60–89%	15 (9.26%)
≥90%	138 (85.19%)
Neutralizing antibodies (%) in neonates	
1–29%	2 (1.23%)
30–59%	6 (3.70%)
60–89%	13 (8.03%)
≥90%	141 (87.04%)

**Table 2 vaccines-11-00620-t002:** Average percentage of neutralizing antibodies detected in the mothers and neonates in relation to the month of vaccination.

Gestational Month of Vaccination	Pairs (Mother–Child)	% Average Percentage of Neutralizing Antibodies in the Mothers	% Average Percentage of Neutralizing Antibodies in the Neonates
1	35 (21.60%)	92%	92%
2	35 (21.60%)	93%	93%
3	35 (21.60%)	88%	88%
4	20 (12.35%)	94%	95%
5	16 (9.88%)	90%	90%
6	10 (6.18%)	96%	96%
7	4 (2.47%)	85%	89%
8	6 (3.70%)	89%	91%
9	1 (0.62%)	98%	98%

**Table 3 vaccines-11-00620-t003:** The average percentage of neonatal and maternal neutralizing antibodies regarding the comparison of the different gestational months of vaccination.

Month of Vaccination Compared	Average 1 Neonate	Average 2 Neonate	Minimum Significant Difference	Months of Vaccination Compared	Average 1 Mother	Average 2 Mother	Minimum Significant Difference
1–2	92%	93%	0.012285714	1–2	92%	93%	0.011142857
1–3	92%	88%	0.042	1–3	92%	88%	0.032285714
1–4	92%	95%	0.030142857	1–4	92%	94%	0.027285714
1–5	92%	90%	0.021482143	1–5	92%	90%	0.018214286
1–6	92%	96%	0.042142857	1–6	92%	96%	0.048285714
1–7	92%	89%	0.035857143	1–7	92%	85%	0.070714286
1–8	92%	91%	0.01252381	1–8	92%	89%	0.025714286
1–9	92%	98%	0.059142857	1–9	92%	98%	0.064285714
2–3	93%	88%	0.054285714	2–3	93%	88%	0.043428571
2–4	93%	95%	0.017857143	2–4	93%	94%	0.016142857
2–5	93%	90%	0.033767857	2–5	93%	90%	0.029357143
2–6	93%	96%	0.029857143	2–6	93%	96%	0.037142857
2–7	93%	89%	0.048142857	2–7	93%	85%	0.081857143
2–8	93%	91%	0.024809524	2–8	93%	89%	0.036857143
2–9	93%	98%	0.046857143	2–9	93%	98%	0.053142857
3–4	88%	95%	0.072142857	3–4	88%	94%	0.059571429
3–5	88%	90%	0.020517857	3–5	88%	90%	0.014071429
3–6	88%	96%	0.084142857	3–6	88%	96%	0.080571429
3–7	88%	89%	0.006142857	3–7	88%	85%	0.038428571
3–8	88%	91%	0.02947619	3–8	88%	89%	0.006571429
3–9	88%	98%	0.101142857	3–9	88%	98%	0.096571429
4–5	95%	90%	0.051625	4–5	94%	90%	0.0455
4–6	95%	96%	0.012	4–6	94%	96%	0.021
4–7	95%	89%	0.066	4–7	94%	85%	0.098
4–8	95%	91%	0.042666667	4–8	94%	89%	0.053
4–9	95%	98%	0.029	4–9	94%	98%	0.037
5–6	90%	96%	0.063625	5–6	90%	96%	0.0665
5–7	90%	89%	0.014375	5–7	90%	85%	0.0525
5–8	90%	91%	0.008958333	5–8	90%	89%	0.0075
5–9	90%	98%	0.080625	5–9	90%	98%	0.0825
6–7	96%	89%	0.078	6–7	96%	85%	0.119
6–8	96%	91%	0.054666667	6–8	96%	89%	0.074
6–9	96%	98%	0.017	6–9	96%	98%	0.037
7–8	89%	91%	0.023333333	7–8	85%	89%	0.045
7–9	89%	98%	0.095	7–9	85%	98%	0.135
8–9	91%	98%	0.071666667	8–9	89%	98%	0.09

## Data Availability

Data will be made available upon request.

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
