# Peer review of "Levels of Neutralizing Antibodies against SARS-CoV-2 in Mothers and Neonates after Vaccination during Pregnancy"

_vaccines, 2023, doi:10.3390/vaccines11030620_

Round 1
Reviewer 1 Report
1. This article focuses on the status of COVID-19 vaccine-induced maternal antibodies. But I did not understand the data and the content of the analysis from the manuscript right now. The author should re-analyze the data and organize the description.
2. I believe there are a few points that may be of interest to readers. For example: for pregnant mothers, how long they should immunize the first dose of vaccine before the expected date of delivery, at least? For pregnant mothers with two doses of the vaccine, are there any differences in the prevalence and persistence of neutralizing antibodies in their children compared with neonates whose mothers received one dose of the vaccine? What effect diet has on antibodies? However, none of these was mentioned in the results in the article. The result data should be further analyzed.
3. Table 3 should have the note. I have no idea about the meaning of the values in each column. For instance, in “months of vaccination”, what is “1-2” meaning? Is it 1st month to N month? Why there are two averages?
4. Table 3. Presenting the most important results in a tabular format is highly unintuitive. Can be rendered using a diagram?
5. Table 3. The number of people used for data analysis is not listed.
6. Discussion. The first paragraph should be deleted.
7. Line 203-204, in “Corroborating with such findings, Popescu et al (2022) The presence of anti-RBD was detected” “The” should be lowercase.
8. The discussion section of this article contains a lot of research-related content, but because it is not particularly condensed, this section is more like a review than a valuable discussion closely linked to the content of this study.
Author Response
Dear reviewer 1.
Thank you for the notes and corrections you requested. I am sure that all of them will be of great contribution for the improvement of the manuscript. Below are the explanations concerning the points requested.
The title of the article has been changed as requested by one of the reviewers, and is now entitled: Levels of neutralizing antibodies against SARS-CoV-2 in mothers and neonates after vaccination during pregnancy
1.
The data were reanalyzed by the statistician who is one of the contributors to the article.
2.
Through the detailed analysis of the data it was possible to arrive at the requested point that is included in the article
The average amount of antibodies in neonates whose mothers received the first dose during gestation (87 mothers) corresponds to 90%; of those, four did not receive their second dose, with an average of 94% of antibodies in the mothers and 88% in the neonates. A total of 66 mothers received two doses, with an antibody average of 89% in mothers and 90% in neonates; eleven received three doses—including the booster dose—with an antibody average of 95% in mothers and 96% in neonates; and six provided no information. From the information above, we can say that in our data, mothers who were vaccinated with a single dose during pregnancy passed on a lower amount of antibodies to their babies than mothers who received two doses, and the same can be seen if we compare mothers who received two doses with those who received three doses.
3 and 4.The table has been modified for better understanding, however, due to the amount of data, a diagram would not be viable to explain it in a didactic way.
5.
The number is listed in table 2
6 and 7.
Corrected items as requested
8.
The discussion has been revised and updated, leaving the most important papers for better contextualization
I thank you for your availability, and look forward to the manuscript being approved and published.
Sincerely
Antônio Oliveira da Silva Filho
Reviewer 2 Report
It would have been interesting to look for antibodies in breast milk
Author Response
Dear Reviewer 2.
Thank you for the contributions attributed to the article. The manuscript has been submitted to MDPI's English proofreading service.
Unfortunately, at this time it was not possible to collect the breast milk.
The title of the article has been changed as requested by one of the reviewers, and is now entitled: Levels of neutralizing antibodies against SARS-CoV-2 in mothers and neonates after vaccination during pregnancy
Sincerely
Antônio Oliveira da Silva Filho
Reviewer 3 Report
1. Please change the title as it is not conveying accurate meaning of study findings
2. Re write METHODS in abstract, there are lot of grammar issues
3. Guthrie test? usually used for PKU. What is the purpose in the current study. Add authenticated reference
4. Manuscript has full of language issues
4. he Center for Disease Control (CDC) e The American College of Obstetricians and Gynecologists (ACOG),.... why "e"
5. against SARS-CoV 0-2?
6. Please mention ethical clearance number
7. Table 1. Mother's age 15-20???, I cannot believe it and the number is 29. How the study has got approved I am not sure
8. Vocabulary should be acceptable for common reader......decreed.......Titling.....etc.
9. Figure 1, in the legend add more information.
Author Response
Thank you for the contributions attributed to the article. The manuscript has been submitted to MDPI's English proofreading service.
The title of the article has been changed as requested by one of the reviewers, and is now entitled: Levels of neutralizing antibodies against SARS-CoV-2 in mothers and neonates after vaccination during pregnancy
All points have been evaluated, the Research Ethics Committee approval number is in the ethical considerations section.
Sincerely
Antonio Oliveira da Silva Filho
Reviewer 4 Report
The article is of great interest, especially beacuse orf the explanation of the importance of vaccination of pregnant women (fo their and a new-borns protection).
The authors supports the recomedatn of vaccinationg of pregnant women.
In my opinion, there is a lack of evaluation between the relationship of the number of dosis of vaccine and antibodies levels. These results would be of great interest because if in the hipothetic case of no difference in the level of antibodies and number of vaccine, there would be sufficient to indicate 1 vaccination dose during the pregnancy and avoid potential side efects of the following doses.
Author Response
Dear Reviewer 4
Thank you for the contributions attributed to the article. The manuscript has been submitted to MDPI's English proofreading service.
The title of the article was changed as requested by one of the reviewers: Levels of neutralizing antibodies against SARS-CoV-2 in mothers and neonates after vaccination during pregnancy
We were able to include the information that was requested. Follows:
The average amount of antibodies in neonates whose mothers received the first dose during gestation (87 mothers) corresponds to 90%; of those, four did not receive their second dose, with an average of 94% of antibodies in the mothers and 88% in the neonates. A total of 66 mothers received two doses, with an antibody average of 89% in mothers and 90% in neonates; eleven received three doses—including the booster dose—with an antibody average of 95% in mothers and 96% in neonates; and six provided no information. From the information above, we can say that in our data, mothers who were vaccinated with a single dose during pregnancy passed on a lower amount of antibodies to their babies than mothers who received two doses, and the same can be seen if we compare mothers who received two doses with those who received three doses
Sincerely
Antonio Oliveira da Silva Filho
Comentado [ED1]: Please check that intended meaning has been retained
Round 2
Reviewer 1 Report
no
Reviewer 3 Report
Authors addressed all the comments satisfactorily